

# Seasonal Snow-Atmosphere Modeling: Let's do it

Dylan Reynolds[1,2], Louis Quéno[1], Michael Lehning[1,2], Mahdi Jafari[1], Justine Berg[3], Tobias Jonas[1], Michael Haugeneder[1,2], and Rebecca Mott[1]

[1]Institute for Snow and Avalanche Research SLF, Davos, Switzerland
[2]School of Architecture, Civil and Environmental Engineering, Ecole Polytechnique Fédérale de Lausanne, Lausanne, Switzerland
[3]Institute of Geography (GIUB) and Oeschger Centre for Climate Change Research (OCCR), University of Bern, Switzerland

**Correspondence:** Dylan Reynolds (dylan.reynolds@slf.ch)

**Abstract.** Mountain snowpack forecasting relies on accurate mass and energy input information to the snowpack. For this reason, coupled snow-atmosphere models, which downscale input fields to the snow model using atmospheric physics, have been developed. These coupled models are often limited in the spatial and temporal extent of their use by computational constraints. In addressing this challenge, we introduce HICARsnow, an intermediate-complexity coupled snow-atmosphere model.

HICARsnow couples two physics-based models of intermediate complexity to enable basin-scale snow and atmospheric modeling at seasonal time scales. To showcase the efficacy and capability of HICARsnow, we present results from its application to a high-elevation basin in the Swiss Alps. The simulated snow depth is compared throughout the snow season to aerial LiDAR data. The model shows reasonable agreement with observations from peak accumulation through late-season melt-out, representing areas of high snow accumulation due to redistribution processes, as well as melt patterns caused by interactions

between radiation and topography. HICARsnow is also found to resolve preferential deposition, with model output suggesting that parameterizations of the process using surface wind fields only may be inappropriate under certain atmospheric conditions. The two-way coupled model also improves surface air temperatures over late-season snow, demonstrating added value for the atmospheric model as well. Differences between observations and model output during the accumulation season indicate a poor representation of redistribution processes away from exposed ridges and steep terrain, and a low-bias in albedo at high eleva-

tions during the ablation season. Overall, HICARsnow shows great promise for applications in operational snow forecasting and studying the representation of snow accumulation and ablation processes.

## 1 Introduction

Patterns in mountainous snowpack are beautifully complex, with sharp cornices contrasted by smooth wind slabs and fresh snow deposits. The process affecting these shapes are equally complicated, comprised mostly of redistribution by wind and

preferential deposition for the aforementioned features (Mott et al., 2018). Wind redistribution acts close to exposed ridges and peaks, where winds have sharp discontinuities in wind speed. This sets up net accumulation and ablation by saltation or suspension. Snowfall itself is modified at the ridge scale via preferential deposition, where some areas of a cross-ridge transect receive more snow than others (Lehning et al., 2008; Zängl, 2008). Preferential deposition has been the subject of focused





research into what mechanisms lead to such deposition patterns. Initial observational studies found that information about
surface winds, either from station data or model simulations, correlated with areas of differential deposition. Lehning et al.
(2008) noted the mechanism of updrafts decreasing the net fall speed of snow particles, while downdrafts would do the opposite.
This should lead to less deposition in the region of updrafts relative to downdrafts, and their study proposed a parameterization
of preferential deposition relating vertical wind speeds to precipitation. Similarly, Dadic et al. (2010); Helbig et al. (2024) found
that higher horizontal wind speeds, as well as vertical wind speeds, correlated to regions of differential snow deposition over an
alpine glacier. Both of these studies explain preferential deposition as a process dependent on interactions between snow and
the near-surface flow field. Mott et al. (2014) challenged this simplified view of the process, observing that interactions between
falling snow and cloud microphysics, mainly via the seeder-feeder mechanism, also played a role in preferential deposition. The
earlier modeling study of Zängl (2008) found a similar mechanism to lead to increased deposition on leeward slopes for solid
hydrometeors. Importantly, this process is expected to occur at elevations more than 100m above the terrain surface. Mott et al.
(2014) also observed that horizontal advection of particles above ridges in the downwind direction played a dominant role in
the process of preferential deposition. A modeling study from Gerber et al. (2019) corroborated these observations, noting that
differences in modeled snowfall along a cross-ridge transect were existent at elevations above 100m above the terrain surface,
suggesting an influence from cloud-microphysical processes. The authors of this study also considered that mean advection
aloft may contribute to this signal, where a peak in precipitation is shifted downwind from over the peak in elevation. Due to
difficulties in separating these two processes when examining final precipitation amounts, Gerber et al. (2019) considered both
processes to contribute to the preferential deposition signal simulated at the 100m above ground level. Notably, the differences
in snowfall at this height explained two-thirds of the surface snowfall differences. Additional modeling by Wang and Huang
(2017) and Comola et al. (2019) supports the conclusion that horizontal advection aloft contributes to preferential deposition.
Viewed together, the basic description of preferential deposition arising from particle-flow interactions remains correct. At the
same time, the notion that it mainly occurs close to the surface, and is thus a direct result of the surface flow field, is uncertain.
The results of Comola et al. (2019) in particular demonstrated that parameterizations of preferential deposition based on surface
measurements are valid only under advection-dominated particle motion.

Atmospheric models are often employed to better consider the processes affecting snow depth patterns in the mountains, as
done by Gerber et al. (2019). These atmospheric models have also been coupled with snow models in a two-way setup (Voor-
dendag et al., 2023; Vionnet et al., 2014; Sharma et al., 2023). Two-way coupling of atmospheric models with snow models
offers benefits to both models. In this configuration, a better representation of the surface snowpack can lead to better estimates
of mass and energy exchanges between the surface and the atmosphere, which then feeds back to the snow model. This has
been found to directly improve estimates of near-surface air temperature and blowing snow sublimation rates (Schlögl et al.,
2018; Groot Zwaaftink et al., 2013). The influence of precipitation on seasonal snowpack during the accumulation season has
already been discussed, while during the ablation season radiation is the primarily driver of changes to the snowpack (Helbig
et al., 2010; Jonas et al., 2020; Mazzotti et al., 2020b). Unfortunately, these two processes are computed by the most expen-
sive parts of modern atmospheric models, the radiation and microphysics schemes. Even more troubling, the heterogeneity
of mountain snowpack is only resolved at horizontal resolutions approaching the hectometer scale and below (Deems et al.,



2006), and this snowpack heterogeneity is precisely what matters for snow hydrological questions (Luce et al., 1998; Lundquist
and Dettinger, 2005). This heterogeneity results from the accumulation processes discussed above, namely preferential deposition and redistribution, as well as fine-scale radiative processes such as shading from cloud cover or terrain. This means that coupled snow-atmosphere models should be run at the hectometer resolution in order to capture hydrologically relevant differences in the snowpack. And, that the two processes which require the most computation time should not be degraded to reduce computational demand.

These conditions have been followed by the earlier studies using coupled snow-atmosphere models in mountainous terrain mentioned above, and as a result these studies have been constrained to simulation periods on the scale of days. This is due to the computational expense of running atmospheric models at such high horizontal resolutions. One exception to this is the usage of snow-atmosphere models over ice sheets, as done by Sharma et al. (2023) with the CRYOWRF model. In this environment the snowpack is found to vary over larger length scales than in mountainous terrain. This is partly due to the lack of terrain obstacles disturbing the wind field, and a homogeneous distribution of snow depth aside from small-scale bedforms (Filhol and Sturm, 2015; Picard et al., 2019). This reduced heterogeneity of snow depth thus permits larger modeling resolutions. Caveat aside, studying the cumulative impacts of dynamic downscaling on mountain snowpack over an entire snow season requires efficient atmospheric models of intermediate complexity.

The snow modeling community has been adopting this strategy, with numerous studies employing a diagnostic wind solver to generate a wind field for simulating wind-driven redistribution (Groot Zwaaftink et al., 2013; Reynolds et al., 2021; Vionnet et al., 2021; Quéno et al., 2023). This efficient approach to generating a 3D wind field can also be implemented within an atmospheric model, as was done in Reynolds et al. (2023) when developing the HICAR model. This creates a computationally efficient atmospheric model capable of providing high-resolution precipitation and radiation data, in addition to a surface wind field required by most intermediate-complexity wind-redistribution schemes. The approach was tested in Berg et al. (2024, in prep), with HICAR downscaling COSMO1 data (www.cosmo-model.org) to force the FSM2trans snow model (Quéno et al., 2023). COSMO1 is a non-hydrostatic atmospheric model which was used to produce operational weather forecasts over Switzerland. Using dynamically downscaled data was found to result in more heterogeneous snowpack than using dynamically downscaled winds alone, better matching the distribution of observed snow depth.

These results motivated the development of a two-way coupled snow-atmosphere model using HICAR and FSM2trans, which will be the focus of this study. Section 2 will discuss how these two models are coupled together and which data they share. Section 3 will present results from the two-way coupled model, focusing on accumulation patterns in complex terrain, the representation of preferential deposition in the model, and lastly the melt patterns. All of these results will be compared to observations of snow depth from aerial LiDAR scans. Finally, these results will be summarized in the last section, with recommendations for future applications and model improvements.





## 2 Methods

### 2.1 Model Coupling

To simulate the seasonal snowpack and processes of snow redistribution in a computationally efficient manner HICAR employs the FSM2trans model (Quéno et al., 2023), which consists of the base Factorial Snow Model 2 oshd variant (FSM2oshd) (Mott et al., 2023; Essery, 2015; Mazzotti et al., 2020a) with additional modules for calculating snow redistribution. This snow model can account for snow accumulation and melt processes as well as redistribution of the snowpack through wind-driven and gravitational transport. HICAR and FSM2trans are coupled in a two-way system, where a static library of FSM2trans routines are integrated into HICAR as the snow module. At each call to the land surface model (LSM) in HICAR, the forcing data required to drive FSM2, including 10 m wind speed, 2 m air temperature and relative humidity, incoming shortwave and longwave radiation components, and precipitation, are supplied by HICAR. In return, FSM2 computes changes to the internal snowpack properties, as well as the sensible heat flux, latent heat flux, and snow surface temperature, which are subsequently utilized by the chain of surface-atmosphere exchange within HICAR. To highlight these model changes and the coupled system's potential for modeling seasonal snow, we refer to the two-way coupled model as HICARsnow in the rest of the study.

Previous validation of HICAR highlighted the need for a more accurate snow model than the one featured in the NoahMP LSM. However, the rest of NoahMP features more rigorous bare-ground and non-snow-covered vegetation dynamics than what is available for non-snow-covered cells in FSM2. To take the best from both LSMs, we run NoahMP at each LSM time step as well. We turn off the internal NoahMP precipitation partitioning when FSM2 is activated, and supply NoahMP with only liquid precipitation from HICAR. When snow falls on a particular grid cell, or if there is already snow on a grid cell, then the results from running FSM2 are used to update that grid cell during a given call to the LSM routines in HICAR. If a cell is snow covered, then FSM2trans simulates its soil physics, while the soil beneath bare cells is handled by NoahMP.

### 2.2 Parallelization of Snow Redistribution

While the original FSM2 snow model only considers local effects of the atmosphere on the snowpack at each grid cell, FSM2trans simulates redistribution, requiring a transfer of information between grid cells. To facilitate this within the parallelization of HICAR, it was necessary to rewrite the redistribution routines used.

Wind-driven redistribution of snow is calculated using the SnowTran-3D scheme (Liston et al., 2007) in FSM2trans. In this scheme, the saltation flux for the local grid cell are first calculated considering the local wind speed, direction, and the surface properties of the snowpack. We note that this saltation scheme has been known to underestimate saltation fluxes (Melo et al., 2023; Doorschot and Lehning, 2002), but it has given reasonable snow deposition patterns in prior studies employing intermediate-complexity snow transport schemes.

The local saltation flux is then considered by summing the local contribution and the flux at the upwind cell. This step requires the use of non-local information, namely from some upwind grid cell. In the non-parallel SnowTran-3D implementation, the operation is simply performed over the whole model domain at once, moving along each cardinal direction. The domain boundary conditions serve as the upwind flux at the boundary grid cells. However, HICAR parallelizes the domain into



a number of discrete images. In the parallel implementation, boundary grid cells on a given image take on the domain boundary condition for the first iteration, and an initial guess for local saltation fluxes is obtained. The saltation flux at the boundary grid

cells of a particular image are then exchanged with boundary grid cells on neighboring images in a standard halo exchange. These updated boundary values are then used to re-run the saltation flux calculation on the local images.

The exchange of boundary estimates of saltation fluxes and re-calculation of local fluxes is then repeated. This approach has been tested with varying numbers of iterations, and an iteration count of 3 was determined to be adequate for computing steady-state fluxes. The methodology is inspired by the approach used in Mower et al. (2023). Once steady-state fluxes are

found, net snow transport and changes to the snowpack properties can be calculated. The same approach is used for calculating transport via suspended snow.

For gravitational redistribution, FSM2trans uses a scheme based on Snowslide (Bernhardt and Schulz, 2010). In this scheme, all grid cells are examined in a given call to the gravitational redistribution module, comparing the local snow depth to a "snow holding depth" which varies for each grid cell according to slope. Grid cells with snow above their snow-holding depth shed

their snow to down-slope grid cells. These down-slope grid cells are then examined for the same condition, with the process repeated until no grid cell has a snow depth greater than its local snow-holding depth. Quéno et al. (2023) added the additional condition that snow holding depth is reduced when a grid cell was passed snow. In this way, the reduction approximates the effect of static or dynamic frictional coefficients when avalanching snow slides. To parallelize this module, Snowslide is run on each image, and any snow found to be sliding "out of" the image is transferred to the neighboring image. This sequence is

repeated an arbitrary number of times to ensure that avalanches are able to run out their full path. Because Snowslide requires a relatively high number of exchanges, and because the exact timing of avalanche release in such a simplified model is not important, the gravitational module is only called once every simulation hour in HICARsnow.

## 2.3 Observational Datasets

This study relies on repeated areal LiDAR surveys of snow depth to validate the snow depth distribution simulated by the

HICAR model. In spring of 2017, three areal LiDAR flights were performed over eastern Switzerland, covering the rugged upper Dischma catchment. The scans include a date near the peak accumulation of snow before the onset of wide spread melt (March 20th), a date 11 days later after warm temperatures and clear skies induced melting of the snowpack (March 31st), and a date in the middle of May, where most snow at lower elevations has melted away. For this May flight, late-season storms have also enriched the snowpack at higher elevations. The area enclosed by these repeat LiDAR flights is shown in Fig. 1 by the

black lines. Part of the upper Dischma catchment is glaciated, making the extraction of snow depth at these locations difficult. This is because movements of the underlying glacier result in shifts of the snow surface, which would be recorded as changes in snow depth by the LiDAR scan. To avoid comparing the model with observations at these locations, glaciated areas have been masked from the LiDAR data and model results using glacier outlines from the Randal Glacier Inventory 6 (RGI Consortium, 2017).

A previous study using HICAR found that the model exaggerated nighttime cooling of the snow surface, and thus 2m air temperature, in the spring (Reynolds et al., 2024, in prep). To compare the ability of previous model versions with HICARsnow,



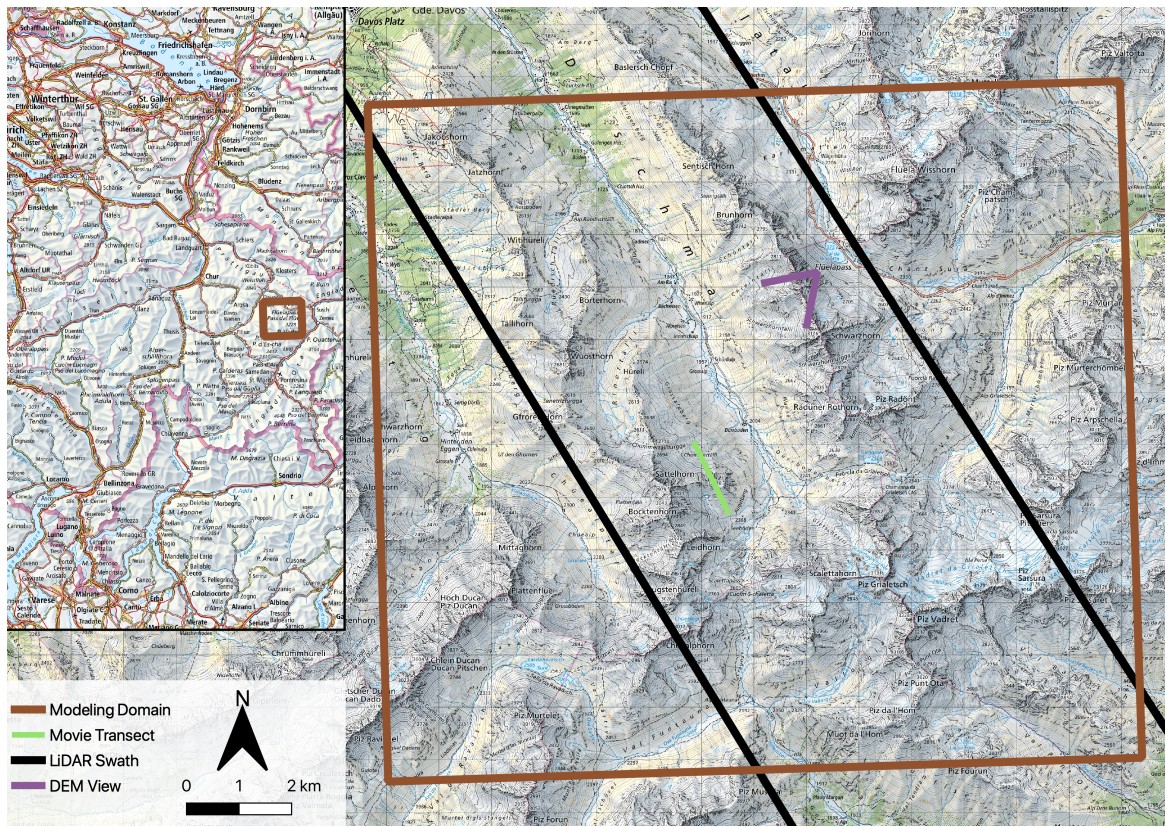

**Figure 1.** Overview of the Upper Dischma Valley outside of Davos, Switzerland (source: swisstopo). The smaller map in the upper left corner shows the location of the zoomed-in plot within the broader Eastern Swiss Alps. The brown box indicates the modeling domain for the 50m horizontal resolution HICARsnow simulations. The black swath indicates the approximate spatial coverage of the LiDAR data introduced in Section 2.3 The green line indicates the location of the transect figures (Figs 6, 7). Lastly, the purple angle shows the viewing angle for the supplementary figure, which compares a 50m and 2m DEM of the region.

2m air temperature data from a ventilated temperature sensor used in this prior study is again used here and discussed in section 3.2. For a full description of the experimental setup used in this earlier study and the conditions present at this time, we refer the reader to the publication.

## 2.4 Model Setup

To test HICARsnow's representation of snow accumulation patterns and snow ablation, the model is run from October 1st 2016 through May 17th, 2017 over the upper Dischma catchment outside of Davos, Switzerland (Figure 1). The simulations are performed at a 50m horizontal resolution, and are one-way nested within larger 250m and 1000m resolution simulations. Topographic data for constructing the Digital Elevation Models (DEMs) is available from the ASTER Global DEM V002 (Spacesystems and Team, 2019), and land surface data from the Corine dataset is used (Agency, 2006). These static data



are then used as input to a domain-generation script distributed with HICAR, which can produce the remaining necessary topographic data. Output from the COSMO1 model was used for meteorological forcing data, including temperature, pressure, water vapor mixing ratio, and the 3D wind field. This data is used to force the outer 1km domain, after which output from the 1km domain simulation is used to force the 250m simulation, and finally 50m. This setup follows that used by previous

studies employing HICAR (Reynolds et al., 2023, 2024, in prep). For the HICAR model, we use version 2.0(TODO: HERE) which features the changes to surface processes detailed in Reynolds et al. (2024, in prep). For the FSM2trans model we use the same model parameters used in Berg et al. (2024, in prep). Of note, FSM2trans can be configured with an arbitrary number of snowpack layers. For this study, we configured the model with 6 snow layers, following the methodology of Quéno et al. (2023). One model simulation was performed over the whole time range with the Morrison microphysics scheme. A shorter

simulation was performed with the ISHMAEL microphysics scheme (Jensen et al., 2017) from October 1st through November 7th 2016 to capture a particular snowfall event. This shorter run was performed due to the nearly doubled model run times when using the ISHMAEL scheme. The ISHMAEL microphysics scheme tracks three forms of ice hydrometeors, or ice "habits", and evolves their density and shape through time to allow for accurate predictions of fall speeds (Harrington et al., 2013a). The scheme belongs to the broader class of Adaptive-habit (AHAB) microphysics schemes (Chen and Lamb, 1994), which have

not yet been employed in the study of preferential deposition. A discussion of the deposition patterns predicted by the two schemes is given in section 3.1.1.

Lastly, in addition to running the two-coupled HICARsnow model, standalone runs using FSM2trans and various forcing data were performed. Two runs with the FSM2trans model were conducted: one run with statistically downscaled COSMO1 data according to Mott et al. (2023) and only the wind field from HICARsnow, and a second run with all of the forcing data provided

by HICARsnow except for precipitation. In this case, precipitation again comes from statistically downscaled COSMO1 data. These two runs are included to demonstrate both the overall impact of dynamic downscaling aside from redistribution, and the effect of dynamically downscaling precipitation alone.

## 3 Results and Discussion

### 3.1 Snow accumulation Processes

Results from running HICARsnow with the Morrison microphysics scheme are shown in Fig. 2, comparing modeled and observed snow depth around peak accumulation. Across the domain, modeled snow depth amounts generally agree with observations, with the valley bottom containing snow depths less than 0.5 m, while higher elevation regions have snow depths near 2 m. Finer scale patterns are also observed in the vicinity of ridges and steep slopes, and these patterns are discussed later in section 3.1.2. Importantly, differences in snow depths exist between the HICARsnow run, and a simulation using FSM2trans

with all of the HICAR forcing data except for precipitation. In this FSM2trans run, we see that there is reduced heterogeneity of snow depth a few hundred meters away from the ridge line compared with the HICARsnow simulation and the LiDAR data. Figure 3 shows that HICAR snow better matches observed snow depth values away from the ridge along a transect bisecting this ridge. Moving to the right towards the ridge, snow depth values steadily increase, and this increase persists after crossing





the ridge before reducing towards the snow depths from the FSM2trans run. This likely arises from the inclusion of preferential

deposition in HICARsnow's precipitation data, and is discussed further in section 3.1.1. From the upper row of Fig. 2 we notice

a bias in HICARsnow towards higher snow depths, particularly on the north-eastern facing slopes near the valley bottom. This

trend is confirmed when binning snow depths according to aspect and elevation, as done in Fig. 4. Here we note excessively

high snow depths along the north-east, east, as well as south-to-west facing slopes. Since this date is near peak accumulation,

and little snow melt has occurred until now, we assume that these snow depth patterns are driven by accumulation processes,

and not melt processes. The higher snow depths in HICAR at lower elevations may be explained by errors in large-scale precip-

itation patterns, such as an incorrect rain-snow line earlier in the season, or an overall wet bias in precipitation, as suggested in

Fig. 5. We also note more evidence of re-distribution processes, such as avalanching or wind-redistribution, at lower elevations

in the LiDAR data than model output. Indeed, HICARsnow mostly represents redistribution processes only in the vicinity of

ridges. Section 3.1.2 will discuss the influence of model resolution on this process representation. In the case of SnowTran-3D,

wind-redistribution is affected by both forcing data from HICAR and the process representation itself, making it difficult to

separate the two as sources of error. At the least, representing redistribution processes at more locations is a clear area of

improvement for FSM2trans, as noted by the original study (Quéno et al., 2023).

Figure 5 shows the effects of dynamical downscaling on snow depth distributions around peak accumulation. As a baseline,

one run is shown where FSM2trans is forced with statistically downscaled output from the COSMO1 model, except for wind

input, which comes from HICAR. This was chosen as the baseline to not include effects of redistribution in the comparison.

The green line then shows the result from including HICAR forcing data for all other variables, except for precipitation. We

note a slight shift to the left, indicating lower snow depths. This may be due a lack of snowfall, as the HICAR temperature field

is now used to partition precipitation into rain or snow, or it may be due to more mid-season melt. The greatest shift can be

seen when using dynamically downscaled precipitation from HICAR. Here, the distribution is both broader, and has a wider

range of values. This result highlights the added value of dynamically downscaling precipitation, where improved gradients

in precipitation result in a broader distribution. Interestingly, the 250m HICARsnow simulation, run without redistribution

processes, shows a similar improvement in snow depth heterogeneity over this domain. Again, we attribute this to the more

heterogeneous precipitation patterns resolved by HICAR relative to the statistically downscaled precipitation input, even at

the 250m scale. Additionally, the HICARsnow simulation at 250m does have a narrower distribution, reflecting the lack of

redistribution processes in the simulation.

### 3.1.1   Snowfall processes

During the accumulation season, snowfall processes shape the pattern of snow depth on the ground, either via orographic pre-

cipitation or preferential deposition. These two processes are most dominant on smooth, flat terrain in the vicinity of ridges. At

these locations, a lack of discontinuities in wind speed driven by terrain features will not lead to net transport via redistribution,

flat terrain will not avalanche, and the proximity to ridges confers a signal of preferential deposition. In Fig. 2 we can see

such a region in the lower panels, comparing the two sides of the dominant ridge. In the LiDAR data we observe deeper snow



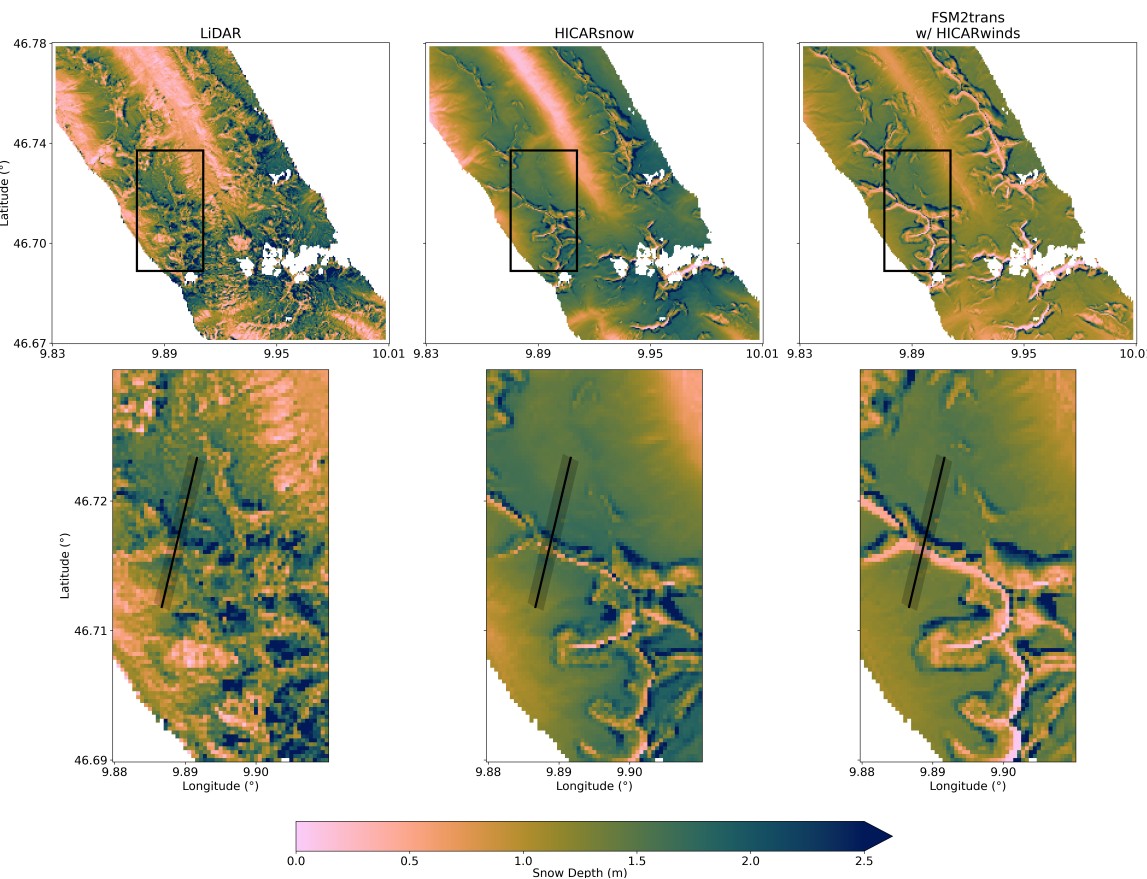

**Figure 2.** Basin-wide comparison of observed snow depth from aerial LiDAR, simulated snow depth from HICARsnow, and simulated snow depth from FSM2trans with all of its forcing data coming from HICAR, except for precipitation. The date is on March 20th, 2017 around peak accumulation of snow. In the lower row, the snow depth around a ridge is shown in detail. The black boxes in the upper row show the location of the detailed view. The model simulations are masked to match the LiDAR flights, where glaciated regions or border cells are removed from the maps.

deposits on the right side of the ridge compared to the left. This general trend is observed in the HICARsnow results as well, but not in the simulation using FSM2trans without dynamically downscaled precipitation.

To better visualize the process of preferential deposition as simulated by the model, two movies of the process have been
made and included in the supplement to this study. Snapshots from two significant moments in the movies are included as Figs. 6 and 7 here. Figure 6 shows the accumulation of snowfall across a ridge during a particular snowfall event on November 6th, 2016. From this event, we observe a clear difference in snow deposition on the windward side versus the leeward side of the ridge. Stronger winds aloft suggest that the dominant process leading to preferential deposition, in this case, is the



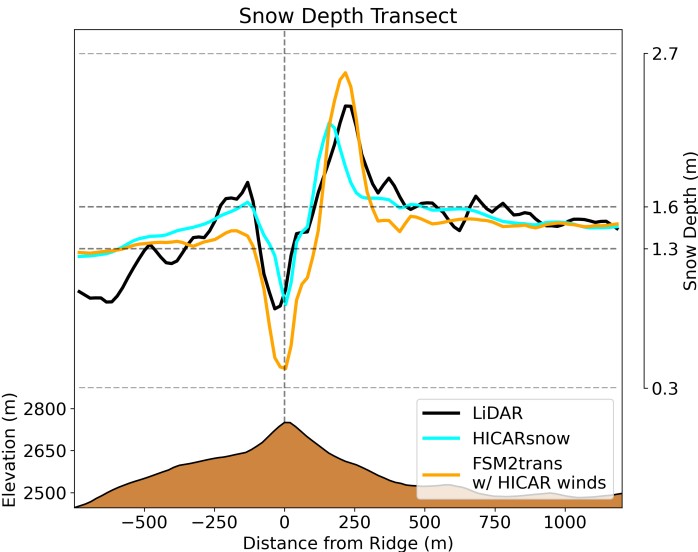

**Figure 3.** Transect of snow depth values, averaged along the transect shown in the cutout of Fig. 2. The direction of the transect is south to north, moving from left to right.

advection of snow particles downstream by winds aloft, resulting in a shift of the peak precipitation distribution (Wang and

Huang, 2017). Interestingly, the snowfall simulated by the two microphysics schemes is roughly similar, with slightly higher snowfall amounts downwind of the ridge in the ISHMAEL simulation than the Morrison simulation. This difference in snowfall amounts is reflected by the higher concentration of snow particles downwind of the ridge in the ISHMAEL simulation. To better grasp why these differences occur, and how the pattern of preferential deposition develops in the first place, a view of the microphysical parameters during this event is presented in Fig. 7.

Here, the complex processes of microphysical interactions, net advection aloft, and near-surface particle-flow interactions are all on display. The ISHMAEL microphysics scheme can track three ice types, planar, columnar, and aggregate ice, and evolve them separately. Aggregates of ice particles were not present during this event, so they are not shown. The Morrison microphysics scheme sorts ice hydrometeors into particular species assumed to have given relationships between particle concentration, mass, and fall speeds. Only snow ice was present in a large concentration for this event. For the initial state

shown in Fig. 7, mean advection aloft is shown to act primarily on planar ice, ice1. The bulge in the distribution of ice1 is shifted downwind in the region of strong horizontal winds. Lower in the atmosphere on the leeward side, the distribution of ice1 has a positive trend, suggesting riming of ice1 as it falls towards the surface. The increase in ice1 fall speeds and positive trend in particle density on the leeward side confirm this. Interestingly, the region of increased fall speeds corresponds to the region of columnar ice, ice2. Ice2 is observed to have much lower particle fall speeds, and thus net fall speeds, than ice1, with

a distribution concentrated around the leeward side of the hill. This suggests that the feeder cloud in the seeder-feeder process





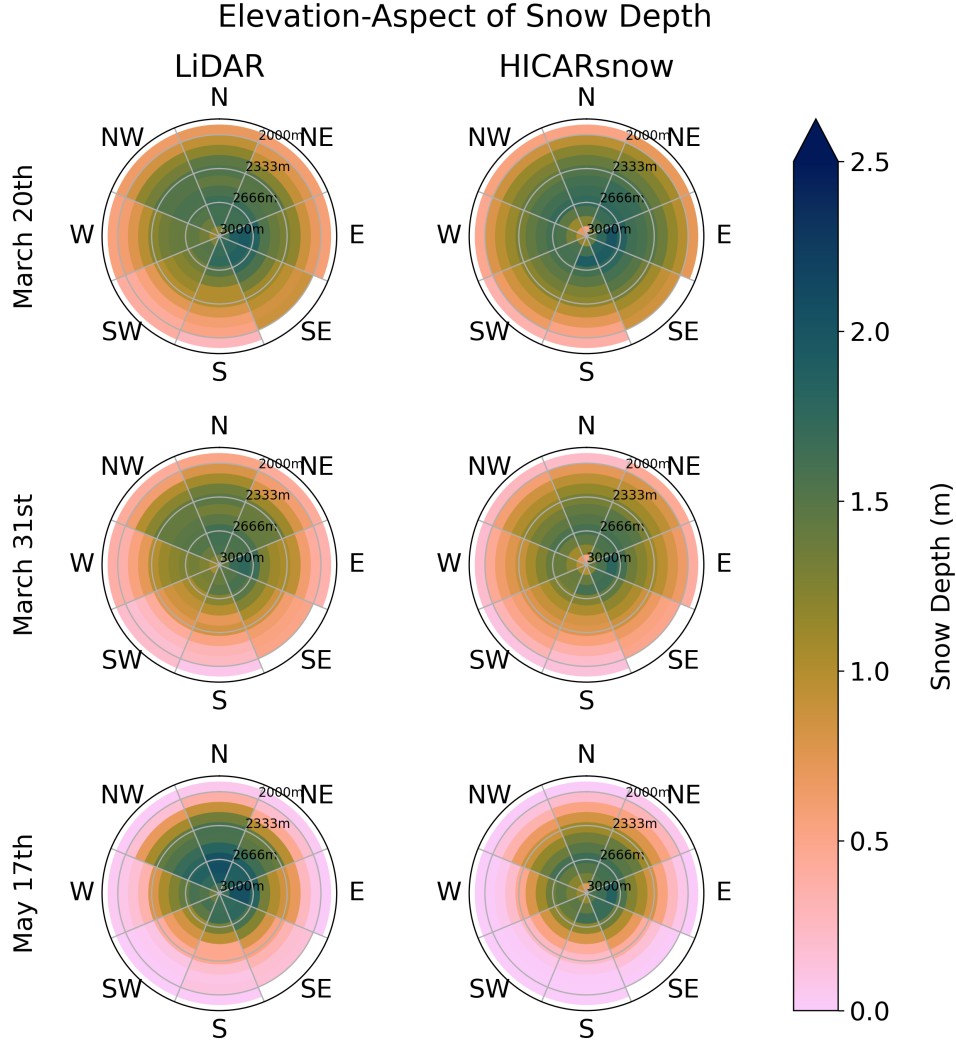

**Figure 4.** Aspect-elevation plots of observed and simulated snow depth above an elevation of 2000m. The data used correspond to the same masked region shown in figure 2.

is shifted downwind of the ridge crest. The cause of this shift is likely a combination of the winds roughly 200 m above the ridge crest, as well as the updrafts present on the leeward side. In this way, we see that cloud-microphysical enhancement via the seeder-feeder mechanism as described by Mott et al. (2014) is also affected by the near-surface flow field. A shifted concentration of snow hydrometeors is also observed for the simulation using the Morrison microphysics scheme, with a bulged





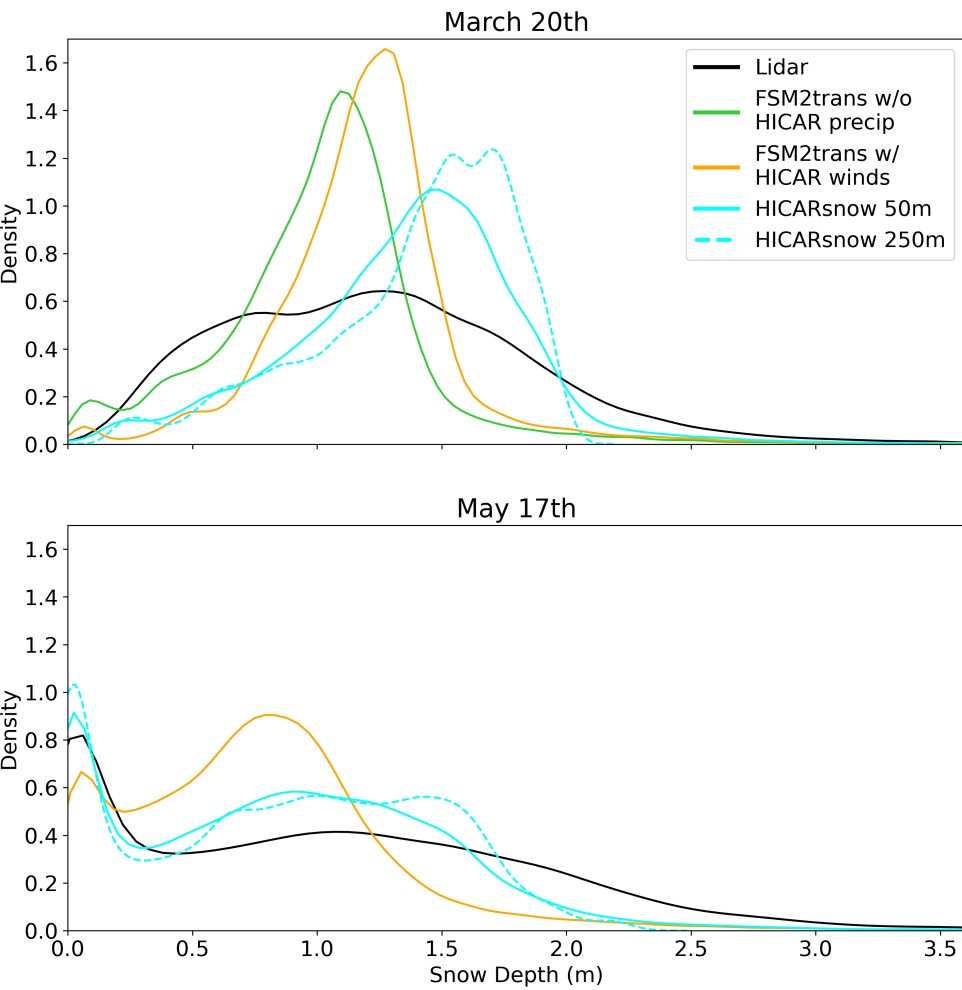

**Figure 5.** Probability Density Functions (PDFs) of observed and simulated snow depth at the times of two LiDAR flights. The orange line shows the results of an FSM2trans run where all of the forcing data comes from statistically downscaled COSMO1 output, except for the winds, which come from HICAR. The green line shows the results of an FSM2trans run where all of the forcing data comes from HICARsnow, except for precipitation, which comes from COSMO1.

distribution of hydrometeors aloft. However, the distribution of particle fall speeds is very homogeneous, indicating that the differences in net fall speed shown by the dashed black contours are mostly due to heterogeneities in the vertical velocity field.



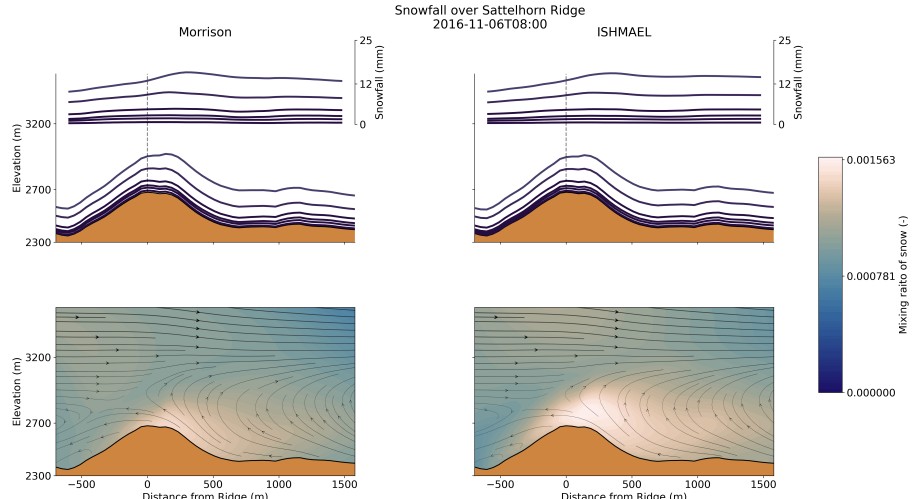

**Figure 6.** Demonstration of preferential deposition during a storm on November 6th, 2016. The location of the transect is shown in Fig. 1, running from the east (left) to the west (right). A vertical dashed line in the upper plot indicates the highest elevation point along the transect. The upper panels show the hourly increase in snowfall (lines in the air) and the hourly increase in snow water equivalent (lines above the terrain) since the beginning of the storm. The lower panels show wind vectors projected along this transect, and the concentration of snow particles in the air. Results using the Morrison microphysics scheme are shown on the left, and the ISHMAEL microphysics scheme is on the right.

The dynamics of this event are best appreciated by referring to the video in the supplement. Over 30 minutes, the feeder cloud in the lee breaks down, and the local water vapor concentration decreases. As a result, fall speeds of both ice species decrease, and their concentrations decline. This leads to a fall-out of the remaining hydrometeors on the leeward side, signaling
the end of this intense period of snowfall. Again, the Morrison microphysics scheme fails to capture these dynamic changes in particle fall speed. It maintains a fairly constant particle fall speed throughout the snowfall event, which has a value similar to the mean fall speed that ISHMAEL predicts for ice1 and ice2 species. The net particle speeds are similar between the Morrison and ISHMAEL simulations, reflecting the importance of the 3D flow field itself in determining sedimentation rates. This similarity likely explains why the deposition patterns shown in Fig. 6 diverge very little. In all, this event was chosen because
it highlights the dynamics that can be simulated with HICAR and shows that the Morrison microphysics scheme produces results consistent with a more detailed, adaptive-habit scheme. Importantly, differences in snowfall patterns between the two schemes do exist, particularly over longer time scales and at spatial resolutions larger than the 50m resolution simulations shown here (Jensen et al., 2018). Still, at these spatial scales, the ISHMAEL scheme simulates more complex microphysical interactions, which give rise to solid precipitation patterns in complex terrain. This comparison also demonstrates the utility of
adaptive-habit microphysics schemes for studying preferential deposition and in particular showing the influence of different types of hydrometeors.



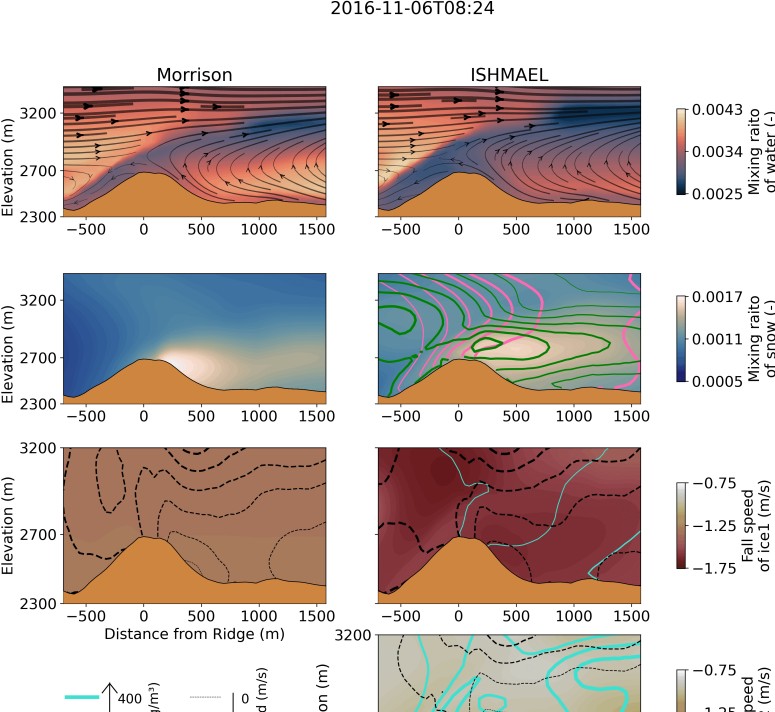

**Figure 7.** The same transect as shown in Fig. 6, but comparing the representation of snowfall in the ISHMAEL and Morrison microphysics schemes. In the upper panels, the wind vectors are overlaid on the water vapor mixing ratio. In the middle panel, the mixing ratio of snow hydrometeors is shown in the background. For the ISHMAEL simulation, the concentrations of ice1 (planar ice; pink) and ice2 (columnar ice; green) are overlaid as contour lines. Thicker lines correspond to higher concentrations. In the lower plots, the ice particles' fall speed are shown as the background shaded color. The hydrometeors' net vertical velocity (vertical air motion - fall speed) is shown with the dashed black lines, where thicker lines indicate faster fall speeds toward the surface. For the ISHMAEL panels, the density of the ice particles is shown by the cyan lines, where thicker lines indicate higher densities. For the ISHMAEL plots, one is shown for the ice1 species, and a second for the ice2 species.

Of note, downdrafts are present on the windward side of the ridge during this event, while updrafts are present on the leeward side. Figure 7 suggests that this is due to eddy-like structures occurring in both valleys which run across the axis of the valley and counter to the mesoscale wind direction. Interestingly, this result contradicts existing parameterizations of 280 preferential deposition based on surface variables (Dadic et al., 2010; Helbig et al., 2024). These previous study identified regions of near-surface updrafts and downdrafts, and correlated them with areas of decreased or increased snow deposition.





In this way, it describes the portion of preferential deposition arising purely from interactions between the near-surface flow and the advection of snow particles. The later studies of Mott et al. (2014) and Gerber et al. (2019) found these particle-flow interactions to be a contributing factor to preferential deposition, but concluded that the interaction between the 3D flow field, snow particles, and cloud microphysics contributed more to preferential deposition of snow.

Synthesizing the results of these studies, we can see that preferential deposition cannot be fully described without knowledge of local cloud microphysical processes and the 3D flow field. This implies that approaches that only utilize 2D near-surface surface fields to parameterize preferential deposition will not capture the dominant effects of a) 3D advection aloft and b) microphysical evolution of snow particles. Earlier models which representing preferential deposition by the advection and diffusion of particles alone will simulate the transport of snow particles (Lehning et al., 2008), but not microphysical processes which may alter their eventual fallout. In the worst case, where local updrafts enhance hydrometeor growth or production, leading to increased fallout, approaches to parameterizing preferential deposition using surface variables will give incorrect results. Since preferential deposition is the dominant process by which precipitation patterns are altered at the hectometer scale, we conclude that dynamic downscaling is necessary to resolve precipitation patterns in complex terrain.

The results of this section demonstrate the complexity of near-surface precipitation processes and the need for dynamic downscaling to capture it. Statistical downscaling is unlikely to capture these precipitation processes which lead to greater variability of snow depths, as shown in Fig. 5. This section has also shown that preferential deposition can occur through interactions between near-surface flow features and microphysical processes. This challenges a dichotomy often invoked when describing preferential deposition (Vionnet et al., 2017), where the two processes shape snowfall distributions independent of each other. Our results suggest that preferential deposition cannot be generally split into two processes, microphysical interactions and near-surface particle-flow interactions, based purely on height above terrain (Gerber et al., 2019).

### 3.1.2 Redistribution processes

In the direct vicinity of ridges and steep terrain, redistribution processes of wind-redistribution and avalanching play a dominant role in shaping the distribution of snow. Importantly, wind-redistribution often feeds the process of avalanching, loading slopes with snow until the weight of the overlaying snow triggers redistribution to lower elevations. In this way, it is difficult to completely disentangle the processes from each other when considering snow depth maps. This is apparent when viewing the cut-out displayed in Fig. 2. Along the ridge line, deep deposits of snow depth are seen to the north of the ridge in the LiDAR data. These deposits may result from wind-loading, avalanching, or a combination of the two. Thus, we will not try to differentiate between the two processes except where obvious and instead focus on the strong heterogeneities present in snow depth around steep or exposed terrain.

Overall, HICARsnow shows good agreement with LiDAR data when representing the heterogeneity of snow depth around the ridges. In particular, the approximate areas of deep deposits are captured well (fig. 2). This results in a weighting of aspect-dependent snow depths at higher elevations (Fig. 4). These observations reflect the findings of Quéno et al. (2023) for FSM2trans in general. Of interest to this study is what patterns of wind-redistribution may say about the wind fields generated by HICAR. One feature of note in the snow depth maps is the lack of wind-redistribution away from prominent terrain features.





Figure 2 displays this, where the secondary ridge found in the upper center of the cut-out features much more heterogeneity in snow depth in the LiDAR data than in the model output. Again, it is difficult to conclude if this results from insufficient wind transport, or avalanching. A similar pattern is seen in the upper right corner of the cutout, where steep, vegetated gullies in the terrain lead to much greater observed snow depth heterogeneity than modeled. This feature is a clue to why redistribution

around secondary ridges is also underrepresented. Both of these terrain features occur over short distances, meaning that they may be poorly represented even in a 50m resolution DEM. Natural disturbances unrelated to the topography (rocks, bushes) should also contribute to increased surface roughness and alter patterns of snow redistribution. The PDF in Fig. 5 shows what effect increased model resolution has on the overall distribution of snow depth, supporting the conclusion that higher model resolutions, or parameterizations which account for sub-grid roughness, may be necessary to resolve snow redistribution in

these areas. The supplementary figure attached shows the difference between representing the domain at a 50m resolution vs. a 2m resolution. This viscerally demonstrates how sub-grid scale topographic features likely alter the wind field at finer scales (Mott and Lehning, 2010), resulting in different patterns of snow depth even when up-scaling to the 50m resolution.

Wind scour of snow is, however, likely overestimated directly at ridges. Figure 2 shows almost 0m of snow depth at some ridge crests throughout the domain, and overall lower snow depth at ridges compared with LiDAR observations. Importantly,

many of these areas of low snow depth are found without corresponding down-slope deposits of snow, ruling out the process of avalanching as a cause of these low snow areas. This excessive scour may be driven by erroneously high wind speeds from HICAR, although a prior study did find reasonable agreement between HICAR's wind speeds and observations at ridge-crests in complex terrain (Reynolds et al., 2024, in prep). Thus, the excessive scour at ridges is likely a combination of high wind speeds and errors an overly simplified relationship between wind speeds and transport in SnowTran-3D. Lastly, we note that

some deep snow deposits present in the LiDAR data exist further from the ridge line than simulated by HICARsnow. These deposits are likely avalanches that have longer run out paths in reality than simulated. Quéno et al. (2023) did address this with a modification to the Snowslide parameterization as discussed in section 2.2, but it may be difficult to accurately represent this process with a simple avalanching model. The issue of model resolution again comes up, where a higher-resolution DEM may represent these slopes at a higher angle, resulting in further run out of the avalanche deposits. Computing wind-redistribution

with a snow-physics model capable of resolving the surface microstructure would also make for an interesting comparison to FSM2Trans. Such a snow-physics model is expected to better estimate the threshold friction velocity, which depends upon the surface microstructure of the snow. Coupling HICAR with such models may also be advantageous, since blowing snow model run times are relatively small compared to atmospheric models (Sharma et al., 2023). Overall, the successes and shortcomings of representing snow redistribution with FSM2trans are in agreement with those of Quéno et al. (2023). We refer the reader to

this publication for a more detailed investigation of the redistribution processes simulated by FSM2trans.

## 3.2   Ablation Processes

Later in the snow season, air temperature and incident solar radiation begin to shape the spatial patterns of snow depth inherited from the accumulation season. The LiDAR data for March 31st in Fig. 4 shows how lower elevations have already begun to experience melt out by this date in the season. Due to the short temporal difference between these two march flights, and the



lack of any precipitation event, the two flights are compared in below to examine HICARsnow's representation of melt patterns. Despite being named the "ablation season", late-season snowfall events can and do occur, as happened between the March 31st and May 17th LiDAR flight. For this reason, the later two flights are not compared for the sake of examining melt patterns. Instead, this flight can be used to test the model's representation of snow depth patterns under a complex situation of melt-out and springtime mixed precipitation events. The PDF of snow depth for May 17th shows good agreement between model output
and observations for very small snow depths 5. These snow depths occur at lower elevations and on southerly slopes at this point in the season, showing that HICARsnow can capture the snow line well compared to observations. This is reflected in Fig 4, where the snow line is found to be within 100m of observations across all aspects. The distribution of snow depths shown in the PDF is close to observations for higher snow depths, but lacks snow depths greater than 160cm when compared to observations. Figure 4 shows that these snow depths occur at higher elevations, and are absent in the model output. The
following paragraph discusses an observed melt bias at higher elevations, which we believe explains this difference in snow depths at higher elevations so late in the season. A prior study comparing HICAR output to observed 2m air temperature also found a slight warm bias when using the Morrison microphysics scheme compared to the ISHMAEL microphysics scheme. A low bias in high-elevation albedo, combined with a slight warm bias, could explain this excessive melt. Thus, future studies using HICARsnow may want to explore using the ISHMAEL microphysics scheme for simulations during the ablation season.
The late-season dry bias at high elevations could also be due to a lack of precipitation, but we have previously noted a slight wet bias in the model results around peak accumulation, and the highest elevations in the domain all experienced solid precipitation events up until May 17th. Thus, we conclude that a bias in the amount of precipitation or phase of precipitation is an unlikely explanation for the dry bias at high elevations on May 17th. Lastly, the 250m resolution HICARsnow simulation shows remarkable similarity to the 50m HICARsnow simulation. This reflects the fact that melt processes largely control the snow depth distribution at this point in the season. Since terrain-dependent radiation parameters such as shading and sky-view fraction are
still included at the 250m simulations, most of this variability in melt patterns is captured even at the 250m resolution.

To visualize the magnitude and spatial distribution of melt processes simulated by HICARsnow, we compare the difference in snow depth from March 20th to March 31st to the observed difference. During this period of March, sunny conditions lead to widespread melt, as observed in the LiDAR data (fig. 8). In particular, snow depth changes occurred mostly at elevations below
2400m, while snow depth patterns above this elevation remained relatively unchanged 4. The maps of change in snow depth also show that southern-facing aspects experienced more melt than northern facing aspects, with this general pattern observed in the model output as well. In particular, smaller scale terrain features, such as the gullies present in the upper right corner of the cutout in Fig 8, show the same pattern of melt as the LiDAR flight. This highlights HICARsnow's ability to simulate slope-scale differences in radiation, thanks in part to the use of terrain-shading factors calculated using the HORAYZON model
(Steger et al., 2022).

While HICARsnow captures the overall pattern and slope-scale differences in snow depth change, the model tends to overestimate melt on south facing slopes, especially at middle elevations. One potential explanation for this is lower albedos predicted by the FSM2oshd model. The albedo scheme used here is a prognostic one, which differs from the scheme used for the operational snow forecast over Switzerland (Mott et al., 2023; Cluzet et al., 2024). The operational scheme was specifically





developed to increase snow albedos at higher elevations. This option was not used for the current study, because it was assumed

that changing the rest of the model forcing data would invalidate the methodology used to tune the operational albedo parame-
terization. Nonetheless, the identification of too low albedos at high elevations in previous studies supports the hypothesis here
that melt on southern aspects is exaggerated due to errors in the snow albedo. Observations from the LiDAR flights also show
a larger decrease in snow depth on south facing slopes relative to northern facing slopes. Solar radiation is an obvious expla-
nation for this difference. Still, the snow difference maps in Fig. 8 show that redistribution of snow onto north-facing aspects
has also enriched snow depths in these locations. Thus, when comparing the same aspects from the HICARsnow simulation,
we can conclude that excessively low snow depths on high-elevation, north facing slopes on March 31st result from a lack of
redistribution onto these slopes.

## 3.3 Snow-Atmosphere Interactions

### 3.3.1 Near Surface Air Temperatures

During the ablation season, temperature, wind, and radiative input dominate the energy input to temperate mountain snow
cover. An earlier study comparing HICAR simulations over spring snow cover found that the model had a large negative
2m air temperature bias during calm, clear nights (Reynolds et al., 2024, in prep). This error was interpreted to be due to
strong radiative cooling of the snowpack. During these calm, stable conditions, excessively low exchange thermal coefficients
are calculated by the NoahMP LSM in HICAR, and thus an uncoupling of the snowpack temperature from the atmosphere
develops. This process has been documented and remedied in other snow modeling studies (Lafaysse et al., 2017; Mott et al.,
2023), and the mechanism of excessively low predicted exchange coefficients has been observed experimentally (Martin and
Lejeune, 1998). Thus, one expectation of coupling HICAR with FSM2 is improving surface air temperatures during such
conditions. To test this, we compare the previous results of Reynolds et al. (2024, in prep) with a run using HICARsnow
for their same modeling setup in Fig. 9. The strong departure from observed temperatures during the night is gone when
using the HICARsnow model, and there is little change in daytime temperature peaks. This result suggests that HICARsnow
can represent diurnal temperature changes during the ablation season, and demonstrates the importance of simulating snow
processes for atmospheric models.

A knock-on effect of this excessive nocturnal cooling is found in Fig. 10. Here, air temperatures in the first model level of
HICAR ( 10m above surface) are shown for just a couple hours after sunrise. We note that the air temperature over snow at
higher elevations is comparable between the HICARsnow simulation and the simulation where HICAR uses the snow model
from NoahMP, albeit with slightly warmer air temperatures in HICARsnow. At lower elevations, however, there are distinct
differences between the two simulations, with the HICAR + NoahMP run producing colder temperatures down slope. This
effect is likely due to the excessive cooling of the snowpack at night producing colder air temperatures across a larger area.
Thus, the excessive radiative cooling at night results in lower air temperatures which persist after sunrise, likely producing less
melt in the HICAR + NoahMP simulation.





**Figure 8.** Maps of snow difference patterns between the two LiDAR flights at the end of March, comparing LiDAR and HICARsnow. Red colors indicate a loss of snow depth from March 20th to March 31st.

### 3.3.2 Blowing Snow Sublimation

Lastly, we can consider the effects of coupling the blowing snow module of FSM2trans, SnowTran-3D, with the HICAR model. Blowing snow, especially via suspension, brings snow crystals into increased contact with the atmosphere where it is possible



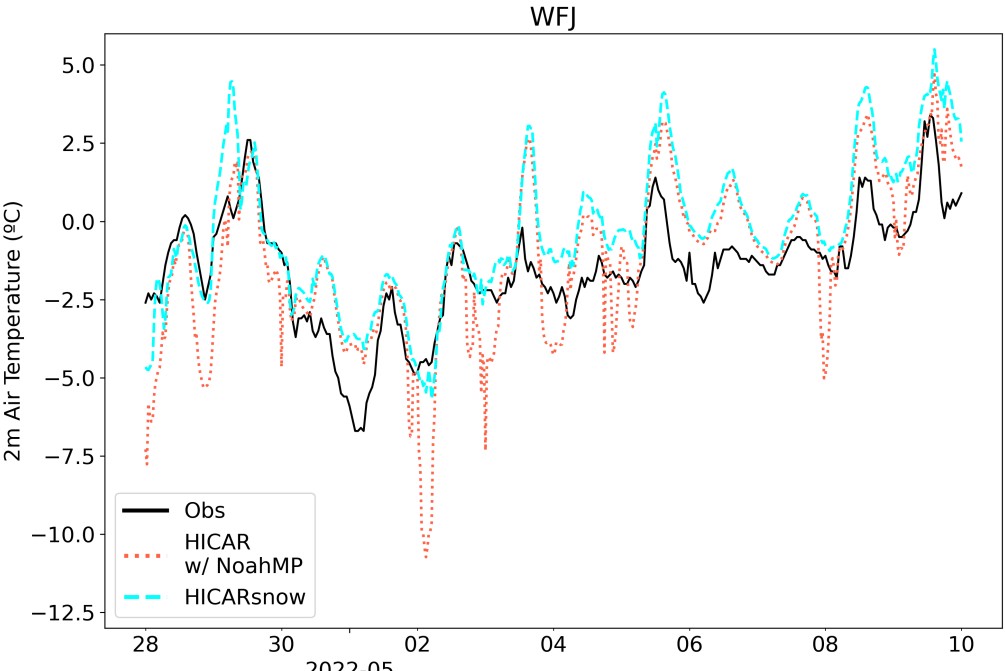

**Figure 9.** Comparison of 2m air temperature from HICAR simulations with observations over a snow covered area in late April 2022. The cyan line shows the results from a simulation with HICARsnow, while the salmon line shows the results from a simulation where HICAR uses the snow model from NoahMP. This figure is a partial reproduction of Figure 2 from Reynolds et al. (2024, in prep).

for them to directly sublimate. If the atmosphere is dry, more sublimation of snow crystals occurs. As sublimation of blowing snow occurs, the surrounding air becomes moist, reducing the efficiency of blowing snow sublimation. This effect has been shown to lead to reduced blowing snow sublimation at the basin scale in alpine catchments (Groot Zwaaftink et al., 2013) and has been documented to completely change surface energy exchange (Sigmund et al., 2022). While studies employing SnowTran-3D have noted basin-wide sublimation rates of  4% of solid precipitation (Bernhardt et al., 2012; Sexstone et al.,

2018; Strasser et al., 2008), the study by Groot Zwaaftink et al. (2013) found a value of only 0.1% when blowing snow sublimation was coupled to the atmosphere. To test this effect on SnowTran-3D, we similarly computed sublimation rates as a percentage of total snowfall over the entire modeling domain shown in Fig. 1. As a result, we find blowing snow to result in just 0.35% mass loss relative to snowfall. In contrast, using just the winds from HICAR to force FSM2trans and the rest of the forcing data coming from statistical downscaling of COSMO1, a rate of 1.2% was found.

These values only reflect domain-wide averages though, and do not speak to the effect that blowing snow sublimation has at individual pixels. Figure 2 shows a considerable difference in snow depths near the ridge crest when comparing HICARsnow versus FSM2 with HICAR winds. In particular, snow depths on the southern facing aspect are lower. When comparing the




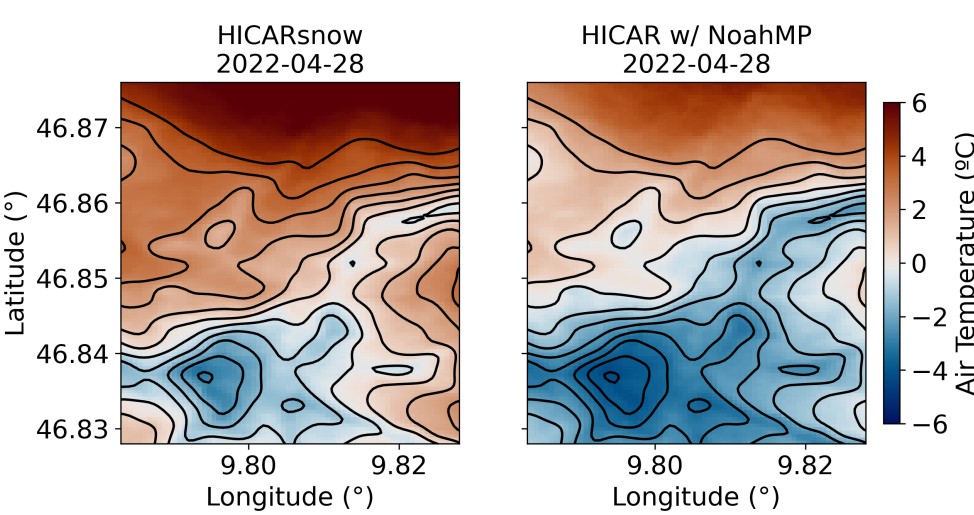

**Figure 10.** Comparison of mean daytime air temperature in the first atmospheric model level for a simulation with HICARsnow, and simulation where HICAR uses the snow model from NoahMP. This covers the same domain and dates as Reynolds et al. (2024, in prep). Black contour lines are shown to represent the topography.

seasonal rates of blowing snow sublimation as a percentage of snowfall at the ridgeline, we see a clear difference between the two simulations (fig. 11). HICARsnow simulates maximum values around 4% at the ridge crest, while the FSM2trans simulation reports values up to 12%. While we do not expect this process alone to not account for all of the difference in snow depth at this ridge, it clearly contributes to the low snow depth values observed in the FSM2trans simulation. This effect demonstrates a clear added value of coupling FSM2trans to HICARsnow, as both models benefit from this improved process representation.

## 4 Conclusions

In this study we have presented results from the first application of a coupled snow-atmosphere model for resolving seasonal snowpack at the hectometer scale. This was achieved by utilizing intermediate complexity snow and atmospheric models which made trade-offs in the representation of some processes in favor of increased run time. To couple the two models together, a static library approach was used, where routines from the FSM2trans snow model are called. Output from FSM2trans is used at snow-covered pixels and replaces the NoahMP LSM at these grid cells. Parallelisation of the snow redistribution schemes





**Figure 11.** Blowing snow sublimation at the ridgeline shown in Fig. 2 as a percentage of local snowfall. Values are cumulative over the entire modeling period, October 1st - May 17th. The FSM2trans run uses only wind data from HICAR, and otherwise uses statistically downscaled COSMO1 output as forcing data.

used in FSM2trans was required, allowing for efficient calculation of snow transport via saltation, suspension, and gravitational redistribution.

Simulated snow depth compares well with snow depth observations from an aerial LiDAR throughout the snow season. Near peak accumulation, spatial patterns in modeled snow depth vary throughout the domain according to aspect, elevation,



and proximity to ridgelines, matching the trend in observations. We attribute this in part to the models ability to represent preferential deposition at the ridge scale, and a particular snowfall event was investigated to disentangle processes that lead to the observed snow distribution. As a result, the basin-wide distribution of simulated snow depth better matches observations than FSM2trans forced without dynamically downscaled precipitation. The coupling of FSM2trans with HICAR shows clear benefits to both models, with FSM2trans improving estimates of near-surface air temperature over snow, especially during clear, calm nights. The ability for blowing snow-sublimation to feedback to the atmospheric model also results in less sublimation overall. This results in an estimate for blowing snow sublimation rates in HICARsnow of 0.35% of annual snowfall. FSM2trans forced with winds from HICAR, but statistically downscaled humidity, estimated this rate to be 1.2%, while another study which considered feedbacks of blowing snow sublimation on humidity reported 0.1% for a similar catchment Groot Zwaaftink et al. (2013).

This study also represents the first time that adaptive habit (AHAB) microphysics schemes have been employed to study preferential deposition. We believe this development to be crucial to further understanding the phenomena, since AHAB schemes promise more physics-based predictions of particle fall speeds. Using this approach, we have demonstrated that the seeder-feeder mechanism involved in preferential deposition can also be impacted by near surface flow regimes. This finding is significant, since existing descriptions of preferential deposition typically bifurcate the process into microphysical processes and near-surface flow (Vionnet et al., 2017; Gerber et al., 2019). Instead, we find that near-surface flow features directly contribute to microphysical processes, indicating that attempts to quantify the relative contribution of each may disregard this unity. However, the conditions during this process would suggest a decrease in precipitation when using parameterizations of preferential deposition based solely on surface-winds. Ultimately, there is a large difference in computational demand between these two approaches to describing preferential deposition, and the use of simpler surface-wind parameterizations may still be useful for some applications under certain atmospheric conditions. In the direct proximity of ridges, snow redistribution is also well represented, with the approximate locations of deposits of snow similar in both LiDAR data and HICARsnow output. These features serve to improve the spatial distribution of snow depths at the basin scale.

Redistribution processes occur mostly at exposed ridges in the model, while observations show evidence of redistribution at secondary ridges and mid-elevation slopes as well. The exact cause of this discrepancy is unclear, but the 50m horizontal model resolution likely plays a role in the ability of the model to represent sub-grid processes generating snow depth variability at these locations (Quéno et al., 2023). Snow melt on southern-facing aspects is also found to be exaggerated by HICARsnow. Prior studies using the FSM2oshd model have found modeled early season melt to be heavily dependent on the snow albedo at these elevations (Cluzet et al., 2024), suggesting that improvements to modeled albedo may yield better maps of early season melt.

In all, this study demonstrates the efficacy of a novel snow-atmosphere model for resolving seasonal patterns of snow depth. The horizontal resolution used for the model simulation controls which processes are capable of being represented, and to what degree of accuracy. This said, coarser resolution (250m) runs with the HICARsnow model yielded distributions of snow depth of similar accuracy to finer resolution (50m) runs with FSM2trans and statistically downscaled data. This suggests that while hectometer-scale HICARsnow runs remain too computationally expensive for operational snow forecasting at the range



scale, some trade off of scale and process representation may be found which rivals higher-resolution approaches. If sufficient
computational resources are available though, coupled snow-atmosphere models show significant promise for representing
seasonal patterns in snow depth at the hectometer scale.

*Code and data availability.* HICARsnow can be used for non-profit purposes under the GPLv3 license (http://www.gnu.org/licenses/gpl-3.0.html, last access: 1 February 2023). The GitHub repository of HICARsnow can be found at:
https://github.com/HICAR-Model/HICAR/tree/HICARsnow with the exact version used for this study found here: 10.5281/zenodo.10679464.
LiDAR data will be made available upon request, and is to be published in full in a later publication. Data from the SMN station are
available at https://opendata.swiss/en/dataset/automatische-meteorologische-bodenmessstationen. The basemap layer used in Figure 1 comes
from Swiss Topo. Similarly, topographic data for generating the DEM resolution comparison was obtained from Swiss Topo swissALTI3d
(https://www.swisstopo.admin.ch/de/hoehenmodell-swissalti3d)

*Author contributions.* DR parallelized the redistribution modules of FSM2trans and coupled the model to HICAR, and ran the HICARsnow
simulations shown here. LQ developed the redistribution modules of FSM2trans, ran the standalone FSM2trans runs, and provided input to
the study design. MJ performed the initial coupling of HICAR with FSM2oshd. JB produced initial, one-way coupled model results which
informed the modeling setup used here. MH, ML, and RM contributed to the experimental setup and advised the direction of analysis. All
authors contributed to the editing of the manuscript.

*Competing interests.* The contact author declares that none of the authors have any competing interests.

*Acknowledgements.* The authors thank the funding source of this project, the Swiss National Science Foundation grant #188554. The computational resources needed to perform the simulations were provided by the Swiss National Supercomputing Center (CSCS) through project
sm78. The authors would like to thank Franziska Gerber for advice on coupled snow-atmosphere modeling setups, as well as additional
developers of the FSM2oshd model including Tobias Jonas, Jan Magnusson, Clare Webster, Giulia Mazzotti, Johanna Malle, and Bertrand
Cluzet. Developers of open source python toolboxes, particularly xarray and xesmf, have also played a crucial role in this study by enabling
efficient analysis and manipulation of large datasets.



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
