# Peer review of "Seasonal Snow-Atmosphere Modeling: Let's do it"

_EGUsphere, 2024_

## Author Comment (AC1)

Section 2.1: This section could be supported with a schematic diagram that outlines the model setup.

We have improved the description of the coupling strategy, namely by adding the sentence "In this way, FSM2trans is coupled to the atmosphere within HICAR similar to how other existing LSM options are." which lays out the coupling more clearly. Following feedback from reviewer 2, the naming of the different coupled runs has also been state more clearly in section 2.1.

Section 2.2: Which of the redistribution processes was calculated first?

Following SnowTran, Saltation is calculated first. This has been made clearer in the schematic figure added.

L 128: How sensitive is the model to different amounts of iterations?

Depending on the prevalence of saltation/suspension, and decomposition of the domain, a single iteration is not sufficient. 2 was found to be sufficient for most circumstances, while 3 was found to be sufficient for very windy, fresh snow conditions using a high level of decomposition. For this reason, 3 was chosen. As mentioned in the text, steady-state fluxes are reached at 3 iterations, indicating that going beyond this does not result in new values calculated.

Section 2.4: How sensitive is the model to the amount of snow layers? Does the top layer represent the surface that interacts with the atmosphere?

The top layer does represent the surface which interacts with the atmosphere. The model is sensitive to the maximum number of snow layers, which higher sensitivity to fewer snow layers. 6 was chosen following the methodology of Quéno et al., 2023. Quéno et al., 2023, which describes FSM2trans, provides a discussion of the layering scheme used in FSM2trans.

L. 170: There is a (TODO: HERE)?

Thank you, this has been fixed.

L. 205: Which criteria are used for partitioning precipitation into snowfall and rainfall? How is the rain-snow slope line defined?

Snowfall and rainfall are partitioned naturally by use of a microphysics scheme. The rain-snow line referenced here is just referring to the elevation where snow transitions to rain – a phenomena which is notoriously difficult to accurately forecast in complex terrain in the months preceding and following winter. This phrase has been changed to "rain-snow transition elevation" in the text to be clearer.

Section 3.1.1: What contributes most to the advection of snow particles and snowfall pattern during the event (wind aloft, above the ridge, or the updraft on the downwind slopes)?

This is tricky to disentangle. How these three flow features affect snowfall deposition involves interdependencies between the individual flow features. For this reason, we have stayed away from commenting on relative contributions (which would likely be very difficult and necessitate arbitrary delineations between the processes), and instead emphasize the interdependency of these different flow patterns. Future studies may examine this by running the model without the lee-side flow parameterization to remove this effect, but such a comparison is outside the scope of the current study.

Section 3.1.2: How much do the two discussed redistribution processes (wind-driven or gravitational redistribution) individually contribute to total redistribution? Can you give an estimate?

This discussion would implicitly pertain to the redistribution models used here, which would repeat the discussion of Quéno et al., 2023. For this reason, a comparison of the relative contribution of these processes is left to the aforementioned study, where it is well discussed.

L. 333: It would be good if there were a brief statement of model performance in terms of meteorological variables earlier in the manuscript.

A relevant sentence referencing the earlier Reynolds et al., 2024 study has been added to the introduction.

L. 355: "5" is "Figure 5"?

Thank you – changed.

L.359: "The following paragraph ..." could be a new paragraph.

Thank you – changed.

L. 363: Is the warm bias due to the too low albedo in high elevations?

How is snow albedo parameterized, and which values are assumed for fresh and accumulated snow?

The warm bias in temperature hypothesized here is attributed to the use of the Morrison microphysics scheme, following the findings of Reynolds et al., 2024. Snow albedo is parameterized using a prognostic scheme, as noted on L384. The value for fresh snow albedo, 0.8, has been added to the text as well.

L. 375: "4" is Figure 4"?

Thank you – changed.

L. 435: Are the differences in the blowing snow sublimation percentage of snowfall due to differences in snowfall or the rate of sublimated snow?

These differences in relative sublimation are due to differences in the rate of sublimated snow – the difference in snowfall amounts between two points in the domain may be a factor of 2 at most, while the relative sublimation reported for the ridge is a factor of 10 greater than the relative sublimation averaged over the whole domain. Additionally, the patterns observed are too spatially heterogeneous to be caused by the snowfall patterns.

---

## Author Comment (AC2)

Reviewer Comment 2

Regarding the general comments, the research questions have been better motivated and laid out. This was done by adding to the introduction a discussion of prior work with the ISHMAEL scheme at coarser spatial resolutions, as well as the hypothesis that better resolving hydrometeor properties which influence fall speed should result in altered deposition patterns relative to two-moment schemes. Some questions are now explicitly given starting on L97.

The in-prep manuscripts which the current modeling strategy is based upon are all open-source, so we do not see that these decisions are "behind" or unavailable to readers at the present time. Some discussion of modeling resolutions necessary for capturing snow depth heterogeneities relevant for snow hydrology is also given starting at L66. That said, we do take the reviewers advice to expand upon the spatial resolutions used in this study, and offer a comment on the topic at L184 when discussing the model setup.

Canopy-wind interactions are indeed an interesting challenge for the model. Aside from the calculation of surface wind speed needed for the land surface model and snow model, the wind field is unaware of surface roughness elements. This could be better included in the future via a parameterization which dampens wind speed over these regions, similar to what was proposed in Reynolds et al., 2023 for regions with negative TPI.

L20 Exposed to what?
Removed

L21 is it really a discontinuity though? The derivative remains well defined. I think this would be better restated as "sharp gradient" or similar
Accepted

L23 "modified" w/c. Modified feels like a model input term
Changed to "altered"

L34 space between amount an dunit – 100m -> 100 m. Fix throughout
Changed throughout

L34 "at elevations" - "at a height above the surface of" might be more clear to match the language on L41

Accepted

L57 "modern" -> contemporary?
Removed

L58 "troubling" w/c
Removed

L65 "These conditions have been followed by the earlier studies" what conditions are these and how can they be followed by an earlier study?
Changed

L66 " This is due to" unclear what this is in reference to
Changed
L67 horizontal and vertical resolutions?
Changed
L67 "One exception to this " what is this?
Changed
L67+ There is a lot of "this" throughout the section, which makes it difficult to follow. Generally I can infer what "this" is in reference to, but it would be best to explicitly state it to ease readability.
Noted, changed
L72 "Caveat aside" which caveat is this in reference to?
Changed
L74 "adopting this strategy" this == intermediate complexity atmospheric models?
Changed
L85 what are the scientific questions, specifically?
Added explicit questions at L97

L90 Later in the MS I found the mix of FSM2trans (e.g. fig 11), HICARsnow, NoahMP, a bit confusing. L 103 notes the name depends on the coupling strategy. I think a small table or a very clear description of all the comparisons would help readability
More clearly described on L199.

L93 define `oshd` ?
Removed

L96 static lib, are you meaning a `.a` ? This is very specific — do you mean this to preclude using a dynamic lib?

Yes, a static library was used with the thinking that this should have less computational overhead than a dynamically linked library.

L109 FSM2 has a soil routine. How are frozen soils + the ground heatflux coupled into this? Or is this fully disabled in this configuration

The soil routine for FSM2 is used for grid cells which are snow covered. Frozen soil and the ground heat flux into the snowpack are considered by FSM2, as described in Essery et al., 2015. Still, the soil model in FSM2, and the corresponding interplay with NoahMP, is a part of HICARsnow ripe for future improvements.

L128 Is there an opportunity to use a dynamic iteration based on an error term versus a fixed iteration count ?

Great question, we wondered this too. In our envisaged implementation, this would require a global reduction of the error term. If one process had reached steady-state fluxes, but an upwind processes had not yet, there may be a need for the current process to re-calculate its fluxes using these neighboring flux values at a future iteration. The inter-process communication required to achieve this likely cancels out the reduction in computations. And, the fixed iteration count chosen has proven to be robust across the seasonal period used here. We considered dynamic iterations to thus be code optimization, and not necessary at this point in the model development.

L125 "image" is not clear. I have a vague recollection this is maybe a iSNOBAL term? For the raster? Can you please clarity.

This is a term from coarrays fortran programming, but has been replaced with "processing element" for more general clarity.

L170 fix TODO: HERE

Removed

L171 can you list, even briefly the details? Difficult to tell with an in-prep manuscript (but I do understand)

Linked to the now published study

L235 Maybe it is just the layout, but it seems that the text goes from citing Fig 2 to Fig 6&7. My recommendation would be number the figures in the order they appear. As is, I'm finding it difficult to find the text that references a specific figure

Figures are numbered in the order they appear – latex references have been used for figure numbers. Agreed that there is a lot of jumping between figures throughout the section though – hopefully typesetting of the final article resolves this.

L 275 Figure 7: What are the purple contours in the 2nd row? I don't see them in the legend. I like this plot though.

There are no purple contours in this plot, but the colored contours in the second row are described in the figure caption.

L331 How is the surface roughness interaction with HICAR modelled?
Surface friction velocity is calculated assuming a log-law transform of wind speeds from the lowest HICAR level and using the surface roughness calculated by the land surface/snow model.

L359 "The following paragraph" — suggest you start a new paragraph
Accepted

L361 how big is "slight"?
Added exact values

L414 Do you think this excessive cooling impact ablation rates? I had originally read this as a FSM2 characteristic, but rereading the text I'm uncertain if it is actually NOAHMP. Please tighten this up a bit
Added a direct reference to NoahMP in this sentence to make it clearer.

L443 these tradeoffs should be clearly noted in the methodology

The caveats of these intermediate modeling approaches have been detailed in prior studies (Reynolds et al., 2023, 2024) and (Liston and Elder 2007, Quéno et al., 2024) and are discussed throughout the manuscript. Still, we mention the two largest shortcomings of their modeling approaches (i.e. turbulent eddies, blowing snow transport) on L460 now.

Citations not in manuscript:

Essery, R.: A factorial snowpack model (FSM 1.0), Geosci. Model Dev., 8, 3867–3876, https://doi.org/10.5194/gmd-8-3867-2015, 2015.